# The Effect of Elevation Gradient on Distribution and Body Size of Carabid Beetles in the Changbaishan Nature Reserve in Northeast Asia

**DOI:** 10.3390/insects15090688

**Published:** 2024-09-11

**Authors:** Shengdong Liu, Jiaqi Tong, Mingfeng Xu, Qingfan Meng, Ying Shi, Hongrui Zhao, Yan Li

**Affiliations:** Jilin Provincial Key Laboratory of Insect Biodiversity and Ecosystem Function of Changbai Mountains, Beihua University, Jilin 132013, China

**Keywords:** indicator species, richness, abundance, tundra

## Abstract

**Simple Summary:**

This study is a first attempt to address the critical issue of effect of elevation on the distribution and body size of carabid beetles in the same region, including mountainous forests and the alpine tundra of the upper limit of vegetation in the north cold temperate zone. The richness, abundance and body size of carabid beetles linearly decrease with elevation increase. The larger carabid species *C. canaliculatus* and *C. venustus* become smaller to adapt to high-elevation tundra conditions. The vegetation type changes at high elevations is an important factor that leads to changes of carabid distribution and body size along the elevation gradient. These high-elevation species such as *C. macleayi*, *N. pektusanica* and *P. jaechi* should be given priority attention in the context of global climate change.

**Abstract:**

The environment of mountain ecosystems can change greatly in short distances as elevation increases. The effects of elevation change on the distribution and body size of carabid beetles were investigated at elevations of 750–2600 m in the Changbaishan Nature Reserve (Northeast China). The richness and abundance of carabid species decreased significantly as elevation increased. However, the change trends are different in forests and tundra. In the broad-leaved Korean pine forest and coniferous forest at low elevations, carabid beetle species have high richness and abundance. The community composition of carabid beetles was significantly different at different elevations and among different vegetation types. Some species only occurred at specific elevations. There were fewer indicator species in high-elevation areas, but *Carabus macleayi* Dejean, *Nebria pektusanica* Horratovich and *Pterostichus jaechi* Kirschenhofer were mainly found in high-elevation areas. The average body size of species in the carabid beetle community was negatively correlated with elevation. The sizes of the larger *Carabus canaliculatus* Adams and *Carabus venustus* Morawitz were negatively correlated with elevation. Their body sizes decreased obviously in the tundra at elevations above 2000 m. Changes in vegetation types at high elevations affect the distribution and body sizes of beetles along the elevation gradient. Some large carabid species may be smaller at high elevations where a unique insect fauna has developed. The body size and distribution range of the carabid may be the factors that affect body size reduction at high elevation. Although some high-elevation species also occur in low-elevation areas, the protection of species diversity in high-elevation areas should be emphasized in the context of global climate change. The results illustrate the mechanisms of carabid beetles’ response to elevation change and the need for carabid beetles’ diversity conservation under global climate change.

## 1. Introduction

Elevation changes affect environmental conditions such as temperature, carbon dioxide levels, ultraviolet radiation and air pressure, which in turn affect insect distribution [1,2,3]. Studies have typically found that insect diversity, richness and abundance decrease with increasing elevation [4]. However, sometimes species richness and abundance reach a peak at mid-elevation [5,6]. Changes in temperature along elevation gradients affect insect population dynamics via physiology responses and behavior, and further affect the distribution range of species [7,8,9]. Different adaptations of different species to temperature can result in unique community compositions of beetles along elevation gradients [10]. Elevation gradients related to species richness is receiving attention as a model system for evaluating ecosystem responses to climate change [11,12]. It has been found that many low-elevation species have started dispersing to high elevations [13], and higher elevations may become a refuge for some species [14,15,16]. Comparative studies of species ecology simultaneously at different elevations can help predict the likely response of species and communities to climate change [17]. Some high-elevation species have disappeared or are at risk of disappearing [18]. More attention should be paid to insect diversity and changes in high-elevation regions.

Carabid beetles live mostly on the soil surface, and are mostly predatory species, and are not confined to a particular host plant or prey [19]. However, they are more sensitive to environmental factors such as falling temperatures than are herbivorous species [20,21]. Aspects of the micro-habitat such as ground temperature and humidity, soil and vegetation cover affect the carabid distribution [21,22]. Like other poikilotherms, carabid beetles develop physiological adaptations to cope with changing environments [23]. Carabid beetles have obvious responses to the change in elevation gradient [19,24]. Some carabids occur in high-elevation habitats at all stages of their life cycle, so they are ideal for testing species responses to changes in elevation [25]. In Europe and other parts of the world, there are also some differences in the elevation gradient distribution of carabid beetles in the mountains [22,24,26,27,28]. These differences in the results may be related to the latitude and elevation range of the study area [29]. Meanwhile, Wolda (1987) suggested that changes in species richness may be related to the duration of collection, and that long-term sampling may show a linear trend, while short-term sampling is more likely to show a peak at a medium elevation [30]. Therefore, reducing the human interference, covering the representative plant types in the study area as much as possible and comprehensive specimen collection are important factors affecting studies on the response of carabids to elevation change.

Body size is an important trait of insects because it is closely related to life cycle, development, reproduction, temperature regulation, viability and dispersal [31,32,33]. Elevation changes can affect not only the distribution of insects, but also the size of insects [34,35]. Insect size often varies along elevation gradients, but there is no consistent pattern in species scale [36,37]. Studies have shown that insect body size is sometimes positively correlated with elevation [38,39]. This is consistent with the pattern expected under the temperature-size rule, which states that body size increases at higher elevations and lower temperatures, which is Bergmann’s rule [40,41]. The main explanation is this when faced with the adverse conditions of limited food resources and low wintering temperatures, larger individuals were able to show higher survival rates [42,43]. However, most studies have shown the opposite pattern [44,45]. Insects are more prone to water loss and often lack the physiological ability to resist dehydration. Smaller individuals may be better adapted to the lack of forest cover and high-elevation light intensity [46,47]. And other studies have found no correlation between insect body size and elevation [48]. Changes in insect body size may also be related to the taxa studied, and mobility, transpiration and dispersal capacity of the adults [23,49]. Some studies have shown that most carabid beetles exhibit the inverse Bergman’s rule [50,51,52]. Studies have also shown that during the larval stage, low temperatures and fewer food resources are important reasons for the smaller adult beetles [53,54]. However, there are significant differences in the degree of body size variation among different species of carabids [55]. The change in body size is a response to the adaptation of species to the change of elevation. Studying the response of the community and the body size of specific species to elevation change will help us understand the adaptation strategies of species to elevation change. This information is of great significance in the conservation of species diversity.

Mountain ecosystems affect insect species distribution through environmental variables associated with elevation. This creates patterns that increase the range of mountain insect niches and increase species diversity [56]. Existing studies mainly focus on Europe and tropical regions, while there are few studies on the response of carabid beetles to elevation change in the northern temperate zone, and there are many knowledge gaps. Because the effect of elevation on beetles may be influenced by the study area, latitude, elevation range, and disturbance, there may be some different results in some regions. Changbaishan is the highest mountain (2749 m asl) in northeast China in the northern cold temperate zone. The Changbaishan Nature Reserve has not suffered from human disturbance such as forest harvest in recent decades, so it is an ideal place to study the effects of elevation gradients on insects. Zou et al. (2014) studied the elevation distribution of the diversity index and abundance in the range of 700–2000 m asl in the Changbaishan [57]. However, the study did not involve the tundra zone above 2000 m asl, the effects of vegetation type changes on the carabid, and the effect of elevation change on the body length of the community. This study area is located in the range of 700 m to 2600 m asl in the Changbaishan, covering the upper limit of vegetation distribution in the north cold temperate zone, including mountainous forests and high-elevation alpine tundra ecosystems with Arctic features.

Our work is the first attempt to reveal the effects of elevation gradients on carabid beetle assemblage in the northern cold temperate zone by studying carabid distribution and body size in the same region. The objective of this study to test (1) Do the richness and abundance of the carabid beetles linearly decrease with elevation increase? (2) Do changes in vegetation types (from forest to tundra) affect the effects of changes in elevation gradient on the carabid beetle assemblages? (3) At the community level and species level, does the body size of carabid beetles decrease with the increase in elevation?

## 2. Materials and Methods

### 2.1. Study Area

The study area was located in the Changbaishan Nature Reserve in Northeast China (127°38′–128°0′ E, 41°42′–42°10′ N) with a surface area of 1965 km^2^ (Figure 1a). The investigated elevations ranged from 720 to 2691 m. The region has a temperate continental monsoon climate. The vegetation landscape ranges from temperate zone to polar. This region has the richest species diversity in the northern hemisphere and is important for biodiversity conservation in Northeast Asia [58]. The study area was on the northern slope of the nature reserve. The vegetation types show a mountain vertical distribution band spectrum with elevation change as follows: (1) The broad-leaved Korean pine forest (720–1100 m) is a large-area natural coniferous and broad-leaved mixed forest. The dominant tree species include *Pinus koraiensis* Sieb. et Zucc., *Tilia amurensis* Rupr., *Fraxinus mandshurica* Rupr., *Acer mono* Maxim, *Quercus mongolica* Fisch. ex Ledeb, *Picea jezoensis* var. *komarovii* (V.Vassil.) Cheng et L.K.Fu. (2) The spruce–fir forest (1100–1800 m) is the main body of forest vegetation on the north slope of Changbaishan with spruce and fir. The dominant tree species include *P. koraiensis* Sieb. et Zucc., *Picea jezoensis* var. *komarovii* (V.Vassil.) Cheng et L.K.Fu, *Picea jezoensis* var. *komarovii*, *Abies nephrolepis* (Trautv.) Maxim. (3) The subalpine birch forest (1800–2000 m) is forest line vegetation dominated by a single *Betula ermanii* Cham. This is the upper limit of forest vertical distribution. (4) The tundra zone (2000–2600 m) has a cold climate with an average annual temperature of −7.4 °C and a frost-free period of 60–70 days. It is a vegetation zone with typical Arctic tundra characteristics with dwarf shrubs. The dominant plant species include *Vaccinium uliginosum* Linn., *Rhododendron confertissimum* Nakai, *Carex pseudo-longerostrata* Chang et Yan, *Dryas octopetala* var. *asiatica* Nakai, *Saussurea tomentosa* Kom (Appendix A).

### 2.2. Experimental Design and Sampling of Carabid Beetles

We set up 16 transects along contour lines from 750 m asl to 2600 m asl on the northern slope of the Changbaishan Nature Reserve. There are three transects in coniferous and broad-leaved mixed forest, four transects in spruce–fir forest, two transects in birch forest and seven transects in alpine tundra. We established four sample plots in each transect, and the distance between the sample plots was about 500 m (Figure 1b). Each sample plot was a circle with a radius of 15 m, and the vegetation was recorded. Specific information on the 16 elevations is shown in Appendix A. A total of nine pitfall trap cups were set up within each sample plot. The pitfall trap spacing was 1 m, and the distribution was three columns and three rows. We used a pitfall plastic cup (diameter = 70 mm, depth = 90 mm) to collect carabids. A small hole was made in the top of the cup to prevent the loss of specimens due to rain. All traps were filled with water and table salt in order to reduce specimen decay.

Carabid beetle collections in 2019 were carried out from 15 May to 1 October. Collections in 2020 were from 15 May to 1 September. The specimens in traps were collected at intervals of approximately 15 d and traps were then supplemented with salt and water. The total collection time in each year covered the activity period of all adult carabid beetles. All carabid specimens were stored in the laboratory of the Forestry College of Beihua University.

### 2.3. Statistical Analyses

To reduce the experimental error of trap catches, we aggregated carabid beetle data from each plot for 2019 and 2020. We used the “ggplot2” package of R 4.1.2 to analyze the Pearson’s correlation of relative elevations and species richness, elevations and abundance, and elevations and average size of species. We used the general linear models to analyze the differences of abundance and richness of carabid beetles in different habits. We performed nonmetric multidimensional scaling (NMDS) visualizations of the distances based on Bray–Curtis dissimilarity to study differences in the composition of carabid beetles in different habitats; we performed 999 permutations of residuals for all tests by the “vegan” and “ggplot 2” package of R (4.1.2). We used the “corrplot” package to perform similarity analysis. This was followed by permutational multivariate analysis of variance (PerMANOVA) to compare the community composition among different habitats. We used the “adonis” function of the “vegan” package R for statistical analysis of the data [59]. To test the response of species to elevation change, we selected the 15 most abundant species and analyzed their effects on elevation increase using the average of four plots in each transect using R for data analysis.

To detect the specificity variation of carabid beetles in the four vegetation types and 16 elevation gradients, indicator species analysis was performed in advance to assess the association of individual beetle species with the 16 elevation gradients [60]. Only species with an indicator value (IndVal) greater than 30, which denoted statistical significance (*p* < 0.05), were assessed by a Monte Carlo randomization test based on 1000 permutations. Indicator species analysis was performed using PC-ORD software (5.0) [61]. To study the effect of elevation change on carabid beetles, the body size of each carabid beetle in each plot was measured. We randomly selected 20 beetle specimens, 10 females and 10 males, and measured their lengths using electronic vernier caliper. If there were fewer than 20 specimens, we measured the size of all specimens collected. We used R to calculate the correlation between body size and 16 elevations.

## 3. Results

### 3.1. Richness and Abundance at Different Elevations

We collected a total of 18,019 carabid beetle specimens, belonging to 62 species. A total of 57 species were collected in the three forest vegetation types, and 26 species were found in the tundra (Appendix A). Among the 16 elevations, species richness and abundance of carabid beetles were negatively correlated with elevation (*p* < 0.001). However, there were differences in the change trend between the forest and tundra. There was no significant correlation between species richness and elevation change in the tundra. Although there was a negative correlation between beetle abundance and elevation in both forest and tundra, the degree of correlation (*r* value) was different in forest and tundra (Figure 2a). These data suggest that the variation of the vegetation type influences the correlation between richness and abundance and elevation. In four vegetation types, species richness decreased significantly with elevation increase (*p <* 0.01), and the abundance in A and B were significantly higher than that in C and in D (*p* < 0.001) (Figure 2b).

### 3.2. Abundance of Species in Different Elevations

Among 15 relatively abundant species, there were differences in their responses to elevation change. *Carabus billbergi* Mannerheim, *Leistus niger* Gebler, *Pterostichus adstrictus* Eschscholtz, *Pterostichus horvatovichi* Kirschenhofer, and *Pterostichus interruptus* (Dejean) are mainly distributed in forests. The abundance peaks of *C. billbergi*, *L. niger*, and *P. adstrictus* were between 750 m asl and 1050 m asl, and the abundance decreased with elevation. *P. interruptus* only occurred below 1800 m asl. *Pterostichus jaechi* Kirschenhofer was found only in the high-elevation tundra, with abundance peaks at 2530 m asl. *Carabus canaliculatus* Adams, *Carabus venustus* Morawitz and *Pterostichus comorus* Jedlicka were distributed at all elevations. The peak of *C. canaliculatus* was at 1200 m asl, the peaks of *C. venustus* were at 1200 m asl and 2080–2180 m asl and the peaks of *P. comorus* were at at 1560–1820 m asl and 2180–2340 m asl. The peak number of *Carabus aurocinctus* Motschulsky was between 1380 m asl and 1670 m asl, and that of *Morphodactyla coreica* (Jedlicka) was between 1560 m asl and 1820 m asl. The peak of *Pterostichus aereipennis* (Solsky) and *Pterostichus pertinax* (Tschitscherine) were at lower elevations between 750 m asl and 1050 m asl, but *P. pertinax* had a small peak at 1820 m asl. *Carabus macleayi* Dejean, *Nebria pektusanica* Horratovich and *P. jaechi* were found at high elevations; *C. macleayi* peaked at 2340 m asl and 2600 m asl, *N. pektusanica* peaked at 2420 m asl and *P. jaechi* peaked at 2530 m asl (Figure 3).

### 3.3. Community Composition

NMDS (nonmetric multidimensional scaling) yielded a two-dimensional solution with a final stress of 0.138. The two-dimensional depiction of these results (based on axes 1 and 2) showed that the carabid beetle assemblage sampled at 16 elevations was compositionally a little distinct from the beetle composition sampled at other elevations. At each of the 16 elevations, the greater the elevational distance, the lower the similarity of beetle community composition. There were larger differences among the four vegetation types at different elevations (Figure 4).

The results of permutational multivariate analysis of variance indicated that the carabids were compositionally distinct among adjacent elevations and four vegetation types (*p* < 0.05), except for B2 and B3, B3 and B4, D3 and D4 (*p* > 0.05). Among the four vegetation types, the compositions of carabid beetle communities were significantly different (*p <* 0.001) (Table 1).

### 3.4. Indicator Species

To detect the intensity of the elevation effect on the distribution of carabid beetles, the indicator species were analyzed in 16 elevations and four vegetation types. Indicator analysis identified a total of 32 species (with individuals > 20, indicator value > 30, *p* < 0.05) (Table 2). Among the 32 species, 13 species were significantly associated with broad-leaved Korean pine forest, 8 species were significantly associated with spruce–fir forest, 3 species were significantly associated with subalpine birch forest, and 5 species were significantly associated with subalpine birch forest in the tundra. In terms of elevation gradient distribution, 9 species were significantly associated with three low elevations in the broad-leaved Korean pine forest, 8 species were significantly associated with four elevations in the spruce–fir forest, 3 species were significantly associated with two elevations in the subalpine birch forest, and 4 species were significantly associated with four elevations in the tundra. *N. pektusanica*, *P. jaechi*, and *C. macleayi* are typical alpine tundra species with high indicator values. In particular, *C. macleayi* was the most abundant at 2600 m, indicating that it is well adapted to the harsh environment present at high elevations.

### 3.5. Body Size of Species

At the 16 elevations, the average body size of carabid beetles was significantly negatively correlated with increased elevation. The body size distribution of 24 indicator species at each elevation shows that the average body size of indicator species at different elevations is also negatively correlated with elevation (Figure 5).

Among the 15 most abundant species, the body sizes of *C. canaliculatus* (*r* = −0.850, *p* < 0.001) and *C. venustus* (*r* = −0.611, *p* < 0.001) showed a significant negative correlation between body size and elevation, and the correlation coefficient was higher. But the correlation coefficient is low between the body size and elevation rise of the two species in the forest (*r* = −0.298, *p* < 0.001; *r* = −0.277, *p* < 0.001) and tundra (*r* = −0.050, *p* = 0.526; *r* = −0.064, *p* = 0.397). The correlation coefficient between body size and elevation was very low for the other 13 species (Figure 6).

## 4. Discussion

### 4.1. Diversity of Species and Elevation

Elevation change not only affects temperature, humidity and other factors in the forest, but also affects vegetation change. The variation in these factors can have a great influence on the species composition, richness and diversity of beetles [62,63]. In this study, there were significant differences in carabid community composition among different elevations and among different vegetation types. The richness and abundance of carabid beetles decreased significantly with an increase in elevation from 750–2600 m in Changbaishan. However, the results are different from those of Zou et al. (2014) [57] in this region. They reported a total of 4834 carabid beetle specimens and 47 species, and the abundance of carabid beetles increased with elevation increase. The difference may be related to the different time of specimen collection (from July to early August, and from late June to late August by Zou et al.) and the different elevation range of study (700–2000 m by Zou et al.). The size of the study area, the capture method, and the amount of available data can lead to different results [64]. Our study is consistent with the results of Maveety et al. (2011), and they found that carabid beetle richness decreased monotonously with elevation, and few species can be found across the entire elevation range in Peruvian cloud forest [27]. The decrease in temperature with elevation may also be a factor in the decrease in the diversity of carabid beetles [65,66]. Meanwhile, at increased elevations, carabid predatory ability may be reduced [67,68]. But Moret (2009) and Maveety et al. (2013) also found that the peaks of carabid beetles were in mid-to-high elevations in the alpine Páramo ecosystem in Ecuador and K’osñipata Valley, in southeastern Peru [22,26], respectively. The species richness and abundance of flightless ground beetles increased with elevation in the tropical rainforests of North-Eastern Australia [28]. There was no significant difference in the richness of carabids between 1860 m asl and 2890 m asl in the Western Dolomites, Italy [22].

In forest, there was a significant decrease in the richness of carabid beetles at nine elevations from 750 to 1920 m, but no decrease in tundra richness at seven elevations from 2080 to 2600 m. Tundra has no tree growth, strong winds and less accumulation of litter on the surface. It differs from forests in vegetation composition and structure and other environmental variables. Under these two environmental conditions, the carabid beetle distribution changes were quite different along elevations, indicating that the changes of vegetation types at high elevations modified the effects of elevation on carabid distribution patterns [69]. This also supports that habitat vegetation type is an important factor driving carabid diversity [29,64]. At the same time, the elevation gradient performance of insects may be different at different elevation scales. The main reason for these differences may be the difference in species distribution patterns due to environmental heterogeneity caused by elevation change [70,71]. In addition, the lack of tree canopy in the tundra appears to be a major factor affecting the diversity of carabid beetles [72].

The vertical distribution of a species is controlled by its environmental tolerance; the maximum population size is reached at an optimal elevation and the population density decreases at elevations above or below the optimal range [17]. Among the 15 most abundant species, there were great differences in the response to changes in elevation gradient, and some species only occurred within a specific elevation range. *C. macleayi*, *N. pektusanica*, and *P. jaechi* were found at high elevations, and *C. billbergi*, *L. niger*, *P. adstrictus*, *P. horvatovichi*, and *P. interruptus* were mainly distributed in forest. This is also related to the adaptability of different species to temperature at different elevations, as well as to different responses to habitat conditions and available resources [73,74]. There are greater numbers of indicator species in the forest at low elevations but fewer indicator species in high-elevation areas. This is consistent with the findings of Zou et al. (2016) showing that the number of specialized species at middle and low elevations is significantly greater than those at high elevations [75]. Although most species occur at lower elevations, some species also maintain small populations at higher elevations. This may be related to the biology of the species and the behavior of actively migrating to higher elevations [29]. However, global climate change could force species to successfully move into higher elevation habitats. The distribution of species populations at different elevation can help evaluate species responses to climate change [9,57].

### 4.2. High-Elevation Species in Alpine Tundra

The tree line was 2000 m asl, and above the tree line was alpine tundra with features of Arctic vegetation in Changbaishan. We found the richness and abundance of carabid beetles to be low in the tundra, and this appears related to the harsh environmental conditions, low temperature, changing weather, lack of vegetation, short growing period for species and poor food resources. These factors affect the survival of species in tundra, which is consistent with the research results on insect diversity in tundra areas [76,77]. Some researchers also believe that the climatic conditions above the tree line at high elevation are the main factors affecting the distribution of carabid beetles [22]. In the seven elevation gradients of the tundra, there was no significant change in the richness of the species, but there was a change in community composition. Although carabid beetle richness was higher at low elevations, there were 26 carabid species living in the tundra above 2000 m asl, indicating that these species have strong adaptability to high elevations. These species may be more sensitive to changes in temperature, not only because they are able to adapt to environmental pressures at high elevations, but also because they may have a lower need for heat and greater tolerance to environment variation [17].

The elevation pattern of species richness may be related to the ecological requirements and life strategies of species [60]. Some studies demonstrate that temperature may also be a diversified selection pressure, and survival in cold conditions at high elevations will eventually lead to unique and specialized alpine species [78,79]. The number of indicator species in the alpine tundra above 2000 m asl was small and included only 4 species. *M. arctica*, *P. jaechi*, *C. macleayi* and *N. pektusanica* are mainly active in the tundra at high elevations, and these species may have physiologically adapted to the temperature changes in the high-elevation habitat [80,81]. The species that have successfully adapted to high elevations are able to cope with the environmental conditions at all stages of their life cycle [13]. We also trapped a small amount of *Cymindis vaporariorum* (Linnaeus) in the tundra, which is also a typical indicator species at high elevations in the European Alps [22]. *C. macleayi* and *N. pektusanica* also have small populations in adjacent low-elevation forests, and the ability of populations to spread or disperse into adjacent low elevations may reduce the risk of population extinction [82]. However, if warming continues, species found only at high elevations could be at risk of extinction. These high-elevation species need protection to prevent their extinction.

### 4.3. Body Size of Species and Elevation

In this study, the sizes of carabid beetles were negatively correlated with the elevation increase. Among indicator species, the species with large-bodied individuals at low elevations were more abundant, while species with small bodies individuals were dominant at high elevations. This is consistent with the findings of Beckers et al. (2020) and Chamberlain et al. (2020), that carabid community body size decreases with elevation increase, mainly due to there being more species with small bodies at high elevations [23,64]. This finding also differed from other reports that demonstrated that high-elevation insects have larger bodies and low-elevation insects have smaller bodies [34,83]. It also suggested that in harsh environments such as light and low temperature at high elevations, smaller insects could have greater adaptive capability [77,84].

Insect growth and development rates differ at different temperatures. This may result in insect body size changing along the elevation gradient [17]. We analyzed 15 species with large collection numbers. Only *C. canaliculatus* (mean body size: 25.37 ± 2.33 mm) and *C. venustus* (mean body size: 19.28 ± 1.26 mm), showed a significant negative correlation with elevation, and the converse Bergmann’s rule. This is similar to research showing that the size of some species decreases with elevation [23,84,85]. This is also consistent with the results of Baranovská et al. (2019) in the Central European Mountains, which show that the carabid body sizes of most species had no obvious response to the change in elevation, while a few species decreased in size with elevation increase [55]. Furthermore, some insect species at high elevations may respond to lower temperatures, shorter growing seasons and limited food resources by reducing their number of instars during development and producing smaller adults [76,86]. Especially in the larval stage, the lack of food resources in the alpine tundra zone is also an important reason for the change in adult body size [53,54].

Some experts also believe that the underlying mechanism of species body reduction is that the time for foraging, growth and development is generally limited, resulting in smaller body size [87]. In this study, the larger-bodied species may not be as adaptive to the environment as the smaller species, and they can only adapt to the change in environmental conditions by adjusting their size. This was not the case for smaller species. The two carabids that exhibited body shrinkage occurred mainly in the tundra above 2000 m asl, which also indicates that changes in vegetation type influenced the effect of elevation on beetle body size. This study also found that *C. canaliculatus* and *C. venustus* are abundant at each elevation and can adapt to environmental conditions there. Whether this means that only species that can be widely distributed at a larger elevation range can become smaller at high elevations remains to be further studied. Meanwhile, *C. aurocinctus*, *M. coreica*, *P. aereipennis* and *P. pertinax*, although widely distributed, are less abundant at high elevations, while *P. comorus* (mean body size: 10.89 ± 0.51 mm) is abundant at all elevations, but its body size is smaller. We think that these factors may also be responsible for their body size not shrinking at high elevations. This information should be paid more attention in future research.

## 5. Conclusions

The research results clearly reflect the influence of elevation change on the distribution pattern of carabid beetle species. The richness and abundance of carabid beetles decreased gradually with increased elevation, and change of vegetation types at high elevations is an important factor that leads to changes of insect distribution and body size along the elevation gradient. The vegetation type changes caused by increased elevation may have a more direct impact on species. Some larger carabid species may become smaller to adapt to high-elevation conditions. The high-elevation region has formed a unique insect fauna, and its community composition differs from that of the low-elevation regions. Some of the species are only adapted to activities in this region. These high-elevation species, such as *C. macleayi*, *N. pektusanica* and *P. jaechi*, should receive more attention. Although some high-elevation beetles may also occur in the low-elevation region, the species in the high-elevation region should be studied to determine the manner in which they cope with global climate changes. This study provides a reference for future studies on altitude-based species diversity conservation during global climate change.

## Figures and Tables

**Figure 1 insects-15-00688-f001:**
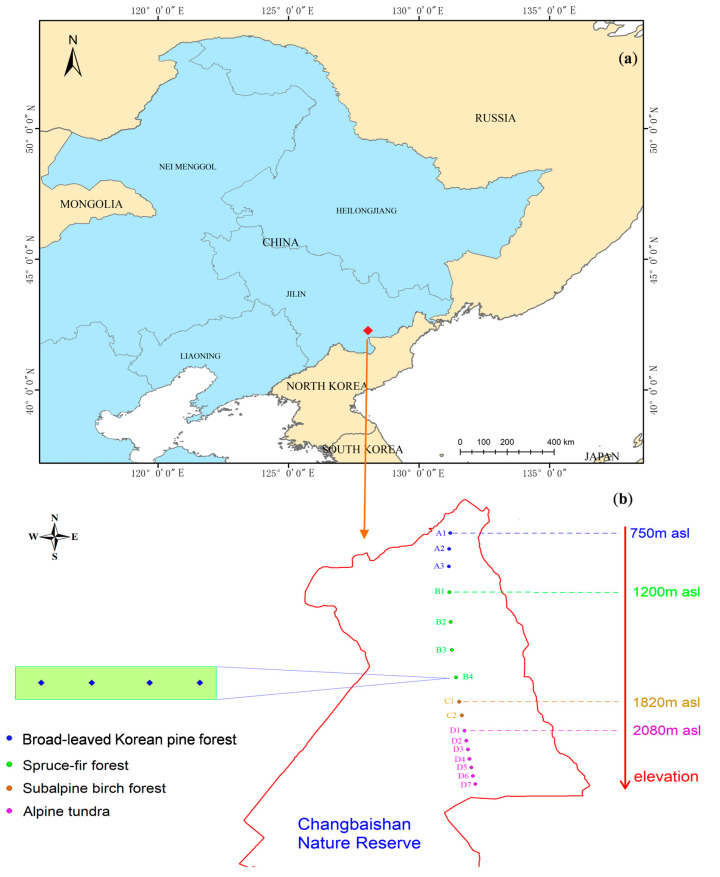
(**a**) The location of study area in Northeast Asia in China. (**b**) The location of 16 transects in different elevations in the Changbaishan Nature Reserve. Circles with different colors represent vegetation types at each transect, and diamond shapes represent 4 sample plots at each transect. A1: 750 m asl, A2: 890 m asl, A3: 1050 m asl, B1: 1200 m asl, B2: 1380 m asl, B3: 1560 m asl, B4: 1670 m asl, C1: 1820 m asl, C2: 1920 m asl, D1: 2080 m asl, D2: 2180 m asl, D3: 2260 m asl, D4: 2340 m asl, D5: 2420 m asl, D6: 2530 m asl, D7: 2600 m asl.

**Figure 2 insects-15-00688-f002:**
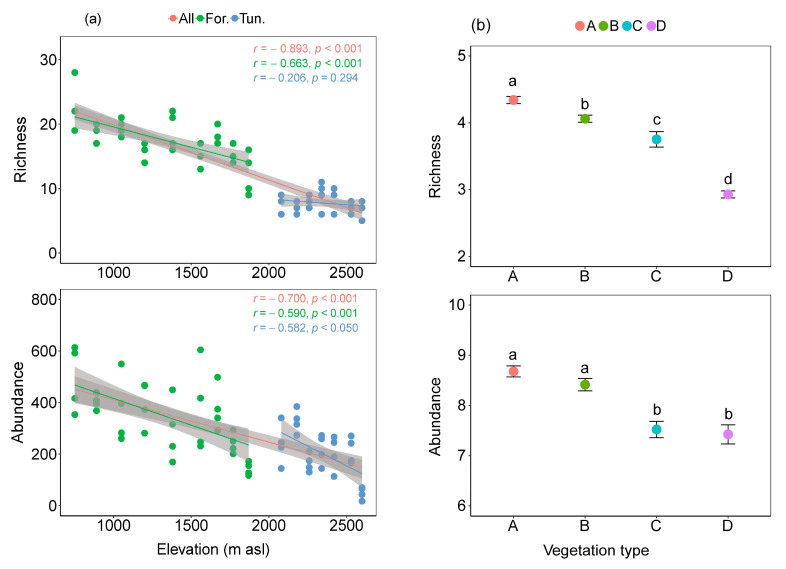
(**a**) Relationships of relative changes in elevations and species richness, abundance, with 95% unconditional confidence intervals (transparent shading) in two years. Each dot represents one sample. The richness indicates the number of species and abundance is the number of individuals collected at each sample. Red represents all 16 elevations, green represents nine elevations of the forest habitat, and blue represents seven elevations of the tundra habitat. (**b**) Richness and abundance of carabid beetles in four vegetation types. Differently marked letters (a, b, c, d) denote significant differences among four vegetation types for richness and abundance (*p* < 0.05). The richness and abundance are the number of species and individuals collected at each vegetation type, and then they were transformed by logarithm base 2. A: broad-leaved Korean pine forest (750–1100 m asl), B: spruce–fir forest (1100–1800 m asl), C: subalpine birch forest (1800–2000 m asl), D: alpine tundra (2000–2600 m asl).

**Figure 3 insects-15-00688-f003:**
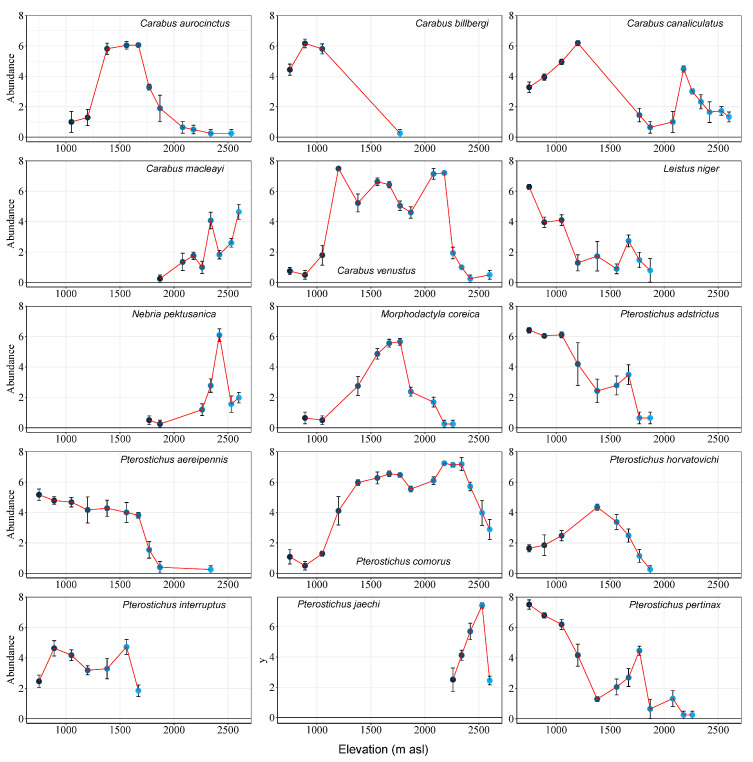
Abundance of 15 carabid beetle species at 16 elevations. The abundance was determined as the number of individuals collected at each sample in two years plus one, transformed by logarithm base 2.

**Figure 4 insects-15-00688-f004:**
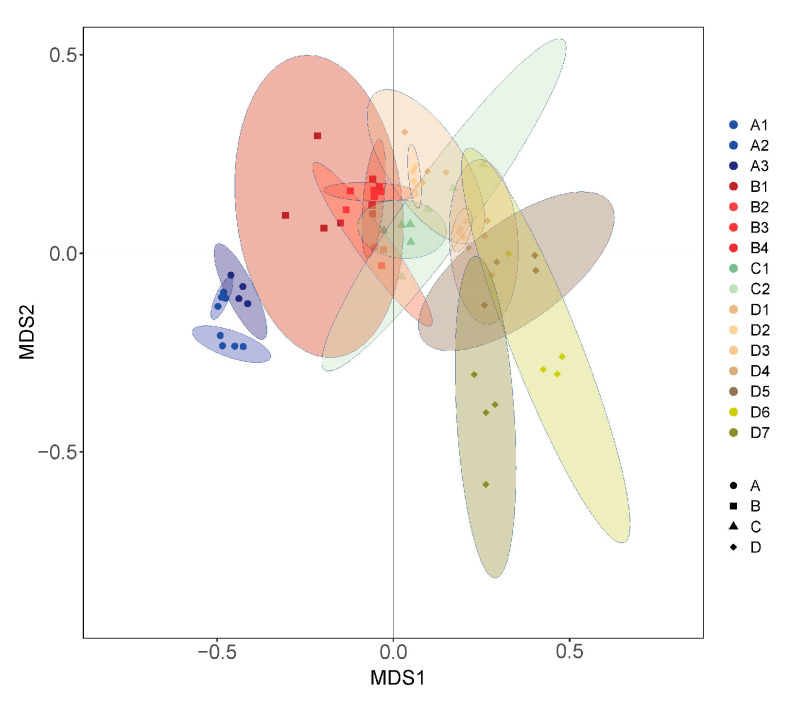
NMDS (nonmetric multidimensional scaling ordination) of carabid beetles at 16 elevations and four forest types represented by different symbols and colors, with 95% unconditional confidence intervals (transparent shading). Each dot represents the data collected at one sample in 2 years, and different colors represent different elevations. A1: 750 m asl, A2: 890 m asl, A3: 1050 m asl, B1: 1200 m asl, B2: 1380 m asl, B3: 1560 m asl, B4: 1670 m asl, C1: 1820 m asl, C2: 1920 m asl, D1: 2080 m asl, D2: 2180 m asl, D3: 2260 m asl, D4: 2340 m asl, D5: 2420 m asl, D6: 2530 m asl, D7: 2600 m asl. A: broad-leaved Korean pine forest (750–1100 m asl), B: spruce–fir forest (1100–1800 m asl), C: subalpine birch forest (1800–2000 m asl), D: alpine tundra (2000–2600 m asl).

**Figure 5 insects-15-00688-f005:**
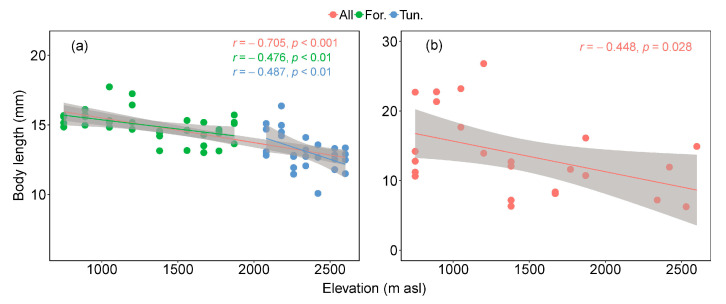
(**a**) Relationships of relative changes in elevations and average size of species, with 95% unconditional confidence intervals (transparent shading). Each dot represents the average size of all species in one sample. Red line represents the entire 16 elevations, green represents nine elevations in the forest habitat, and blue represents seven elevations in the tundra habitat. (**b**) Relationships of relative changes in elevations and average size of 24 indicator species, with 95% unconditional confidence intervals (transparent shading). Each dot represents the average size of one indicator species.

**Figure 6 insects-15-00688-f006:**
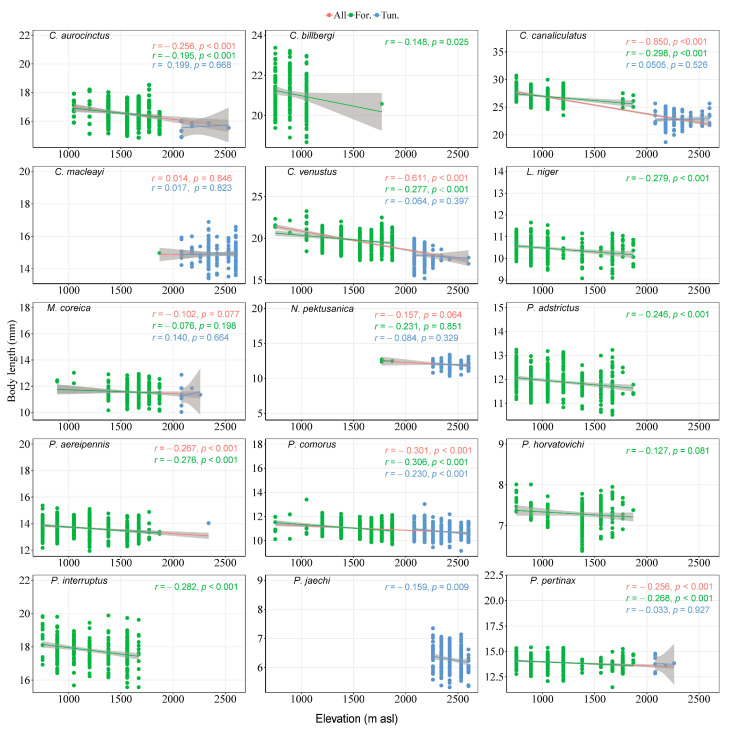
Relationships of relative changes in elevations and average size of 15 species, with 95% unconditional confidence intervals (transparent shading). Each dot represents the size of every species in one sample. Red line represents the entire 16 elevations, green represents nine elevations in the forest habitat, and blue represents seven elevations in the tundra habitat.

**Table 1 insects-15-00688-t001:** Results of permutational multivariate analysis of variance of carabid beetle assemblages at different elevations. “*” indicates significant difference between the positions (*p* < 0.05). A1: 750 m asl, A2: 890 m asl, A3: 1050 m asl, B1: 1200 m asl, B2: 1380 m asl, B3: 1560 m asl, B4: 1670 m asl, C1: 1820 m asl, C2: 1920 m asl, D1: 2080 m asl, D2: 2180 m asl, D3: 2260 m asl, D4: 2340 m asl, D5: 2420 m asl, D6: 2530 m asl, D7: 2600 m asl. A: broad-leaved Korean pine forest (750–1100 m asl), B: spruce–fir forest (1100–1800 m asl), C: subalpine birch forest (1800–2000 m asl), D: alpine tundra (2000–2600 m asl).

Adjacent Elevation/Vegetation Type	*R* ^2^	*p*-Value	Adjacent Elevation/Vegetation Type	*R* ^2^	*p*-Value
A1 vs. A2	0.539	0.029 *	D3 vs. D4	0.268	0.124
A2 vs. A3	0.244	0.035 *	D4 vs. D5	0.538	0.038 *
A3 vs. B1	0.796	0.027 *	D5 vs. D6	0.654	0.028 *
B1 vs. B2	0.724	0.039 *	D6 vs. D7	0.744	0.033 *
B2 vs. B3	0.295	0.118	A vs. B	0.591	<0.001 *
B3 vs. B4	0.190	0.271	A vs. C	0.711	<0.001 *
B4 vs. C1	0.571	0.024 *	A vs. D	0.436	<0.001 *
C1 vs. C2	0.718	0.024 *	B vs. C	0.303	<0.001 *
C2 vs. D1	0.699	0.020 *	B vs. D	0.236	<0.001 *
D1 vs. D2	0.403	0.029 *	C vs. D	0.159	<0.001 *
D2 vs. D3	0.794	0.027 *			

**Table 2 insects-15-00688-t002:** Carabid beetle indicator species analysis in 16 elevations and four vegetation types. IndVal = indicator value, significant difference (*p* < 0.05). A: broad-leaved Korean pine forest (750–1100 m asl), B: spruce–fir forest (1100–1800 m asl), C: subalpine birch forest (1800–2000 m asl), D: alpine tundra (2000–2600 m asl).

Species	Elevation Gradient			Vegetation Type		
	Elevation (m)	IndVal	*p*-Value	Vegetation Type	IndVal	*p*-Value
*Carabus granulatus* Linnaeus	750	76.9	<0.001	A	66.7	<0.001
*Leistus niger* Gebler	750	59.8	<0.001	A	86.7	<0.001
*Pristosia proxima* Morawitz	750	54.3	<0.010	A	64.3	<0.010
*Pterostichus pertinax* (Tschitscherine)	750	43.2	<0.001	A	85.5	<0.001
*Pterostichus nigrita* (Paykull)	750	68.2	<0.010	–	–	–
*Carabus billbergi* Mannerheim	890	47.6	<0.010	A	99.8	<0.001
*Carabus wulffiusi* Morawitz	890	52.5	<0.010	A	96.8	<0.001
*Carabus fraterculus* Reitter	1050	34.1	<0.010	A	96.3	<0.001
*Carabus seishinensis* Lapouge	1050	49.5	<0.001	A	58.2	<0.001
*Pterostichus adstrictus* Eschscholtz	–	–	–	A	80.7	<0.001
*Pterostichus aereipennis* (Solsky)	–	–	–	A	58.2	<0.001
*Pterostichus eobius* (Tschitscherine)	–	–	–	A	56.1	<0.001
*Pterostichus gibbicollis* (Mostschulsky)	–	–	–	A	55.7	<0.001
*Pterostichus interruptus* (Dejean)	–	–	–	A	55.7	<0.001
*Carabus canaliculatus* Adams	1200	41.9	<0.001	–	–	–
*Pterostichus tuberculiger* (Tschitscherine)	1200	88.5	<0.001	–	–	–
*Cychrus morawitzi koltzei* Roeschke	1380	55.2	<0.001	B	52.4	<0.001
*Notiophilus aquaticus* (Linnaeus)	1380	48.1	<0.010	B	34.4	<0.050
*Pristosia vigil* Tschistcherine	1380	45.9	<0.010	B	60.6	<0.001
*Pterostichus horvatovichi* Kirschenhofer	1380	40.6	<0.001	B	50.1	<0.010
*Leistus janae* Farkac *et* Plutenko	1670	61.0	<0.010	B	37.4	<0.010
*Peiyuia* sp.	1670	41.4	<0.001	B	42.4	<0.010
*Carabus aurocinctus* Motschulsky	–	–	–	B	80.6	<0.001
*Carabus venustus* Morawitz	–	–	–	B	58.2	<0.001
*Morphodactyla coreica* (Jedlicka)	1770	35.1	<0.010	C	55.5	<0.001
*Pterostichus microps* Heyden	1870	51.6	<0.001	C	80.4	<0.001
*Xestagonum elytroplanum* Morvan	1870	93.1	<0.001	C	85.4	<0.001
*Miscodera arctica* Paykull	2340	38.5	<0.050	D	42.9	<0.010
*Nebria pektusanica* Horratovich	2420	83.8	<0.001	D	62.5	<0.001
*Pterostichus jaechi* Kirschenhofer	2530	65.0	<0.001	D	71.4	<0.001
*Carabus macleayi* Dejean	2600	45.4	<0.001	D	91.6	<0.001
*Pterostichus comorus* Jedlicka	–	–	–	D	39.1	<0.050

## Data Availability

The data that supports the findings of this study are available in the Appendix A of this article.

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
