# Peer review of "The Effect of Elevation Gradient on Distribution and Body Size of Carabid Beetles in the Changbaishan Nature Reserve in Northeast Asia"

_insects, 2024, doi:10.3390/insects15090688_

Round 1

Reviewer 1 Report

Comments and Suggestions for Authors

Title: Delete “(Col ~ dae)” because “carabid beetles” mean same thing as Carabidae of Coleoptera.

Abstract: L35-36. Add “,” between “low-elevation area” and “the protection of“

Keywords: Use other keywords instead of “carabid beetle”, “elevation”, “distribution”, and “body size”, because those keywords are already presented in the title.

Please check font and its size throughout ms.

Introduction: More importantly, please revise the Introduction section carefully, especially in first three paragraphs. Those are just history of previous studies but not included your findings in terms of study background and study aims. In this aspect, although there were so many citations, but I couldn’t find key topic what you want to tell readers. In my opinion, fourth paragraph is only and relatively well described the study background. Each paragraph is focusing elevation gradient, carabids, body sizes, and mountain ecosystems, but I feel that those subjects are not written logically. Please re-construct and revise. Study aims in L117-124 should be separated from main body of fourth paragraph.

L43-45. First two sentences are meaning similar things, so revise please.

L132. [62] -> why this citation is bold and blue character? Please check throughout the ms.

L134-140. Add the scientific name of dominant plants for each vegetation type.

L175. R ver. 4.1.2 (Citation) to ~, delete software

L182. Delete “of R software”, in addition, some sentences are redundant, so you have to need revise the methods using R.

L188. “native” package?

L207. richness -> species richness

L209. Is this correlation coefficient derived from Pearson’s or Spearman’s correlation? If it is, I can’t find any relevant sentence in Materials and Methods. In addition, please use lowercase r instead of R.

L209-211. Discussion. This section is for Results.

L248. Use NMDS

L262-263. This is partly true. In my opinion, vegetation is the other important factor in addition to elevational distance between study elevations (see Table 1). I think the similarity analysis in Fig. 4 may have low potential to understand carabid communities across elevational gradient because it was also displayed in Fig. 4a and readers can be inferred from it.

L276-285. Current form is good, but can you add IndVal for four different vegetations? It has the other potential to understand carabid communities.

L401-413. I agree that tundra is harsh environment, but it means that there are no trees providing stable habitats for large-bodied carabids. Wing loads, thin air, reduced oxygen and so on could be affected carabid distribution. However, sunlight without shading area is more important factor in my opinion, so tundra is like open-habitat in lower elevation. Open-habitats are generally frequently disturbed in terms of temperature and soil moisture that could be influenced by sunlight. Please carefully revise this paragraph.

L415. elevation gradien -> elevational gradient

L452. Insect flora -> Insect fauna

Comments on the Quality of English Language

I found errors and typos in English grammar. I recommend you to have your English proofread.

Author Response

Comments 1: Title: Delete “(Col ~ dae)” because “carabid beetles” mean same thing as Carabidae of Coleoptera.

Response1: We are very grateful to the expert for numerous meticulous and professional revision of the manuscript. It took him a lot of time and energy. We have carefully revised the paper according to the expert's opinion. Make the expression of the paper more clear. Changes made in the manuscript are highlighted in red. The author has answered the reviewer' comments one by one. The point-by-point response to the comments are highlighted in blue italics.

According to the expert's opinion, we deleted “(Coleoptera: Carabidae)”.

Comments 2: Abstract: L35-36. Add “,” between “low-elevation area” and “the protection of“.

Response2: According to the expert's opinion, we added “,” between “low-elevation area” and “the protection of“.

Comments 3:  Keywords: Use other keywords instead of “carabid beetle”, “elevation”, “distribution”, and “body size”, because those keywords are already presented in the title.

Response3: According to the expert's opinion, we replaced keywordscarabid beetle”, “elevation”, “distribution”, and “body size” with “indicator species”, “richness”, “abundance”, and “forest”.

Comments 4: Please check font and its size throughout ms.

Response4: We are very grateful for the expert opinion, we have carried out a comprehensive review of the format and font size of the paper.

Comments 5:  Introduction: More importantly, please revise the Introduction section carefully, especially in first three paragraphs. Those are just history of previous studies but not included your findings in terms of study background and study aims. In this aspect, although there were so many citations, but I couldn’t find key topic what you want to tell readers. In my opinion, fourth paragraph is only and relatively well described the study background. Each paragraph is focusing elevation gradient, carabids, body sizes, and mountain ecosystems, but I feel that those subjects are not written logically. Please re-construct and revise.

Response5: We also agree with the experts. Indeed, the first three paragraphs are only a description of the current situation of the research, but do not provide a systematic summary, nor do they introduce the problems to be focused on in this study. In response to the opinions of experts, we have made a relatively major revision to the introduction. Make the introduction structure more reasonable.

Comments 6:  Study aims in L117-124 should be separated from main body of fourth paragraph.

Response 6: According to the expert's opinion, we separated aims from main body of fourth paragraph.

Comments 7:  L43-45. First two sentences are meaning similar things, so revise please.

Response7: According to the expert's opinion, we recombine these two similar sentences to make their meaning more clearer.

Comments 8:  L132. [62] -> why this citation is bold and blue character? Please check throughout the ms.

Response8: We are very grateful for the expert opinions, we have carried out a comprehensive review of the format and font size of the paper including citation.

Comments 9:  L134-140. Add the scientific name of dominant plants for each vegetation type.

Response 9: According to the expert's opinion, we added scientific name of dominant plants for each vegetation type.

Comments 10:  L175. R ver. 4.1.2 (Citation) to ~, delete software

Response 10: According to the expert's opinion, we deleted “software” after “R”.

Comments 11:  L182. Delete “of R software”, in addition, some sentences are redundant, so you have to need revise the methods using R.

Response11: According to the expert's opinion, we deleted “of R software”. We also revised the description of some sentences in this paragraph.

Comments 12:  L188. “native” package?

Response 12: We deleted the ambiguous word "native" from the sentence.

Comments 13:  L207. richness -> species richness.

Response13: According to the expert's opinion, we added “species” before “richness”.

Comments 14: L209. Is this correlation coefficient derived from Pearson’s or Spearman’s correlation? If it is, I can’t find any relevant sentence in Materials and Methods. In addition, please use lowercase r instead of R.

Response 14: It is Pearson’s correlation, and I added these in Materials and Methods. We use “r” instead “R”in the manuscript, including in the figure.

Comments 15:  L209-211. Discussion. This section is for Results.

Response 15: According to the expert's opinion, we deleted “These data suggest that the variation of the vegetation type influences the correlation between richness and abundance and elevation.” in Results.

Comments 16:  L248. Use NMDS.

Response 16: According to the expert's opinion, we added “NMDS” in this sentence.

Comments 17:  L262-263. This is partly true. In my opinion, vegetation is the other important factor in addition to elevational distance between study elevations (see Table 1). I think the similarity analysis in Fig. 4 may have low potential to understand carabid communities across elevational gradient because it was also displayed in Fig. 4a and readers can be inferred from it.

Response 17: We fully agree with the expert that there is indeed some duplication between Figure 4a and Figure 4b. and the original purpose of using Figure 4a and Figure 4b is to better present the results. Since the contents in Table 1 can also reflect the contents in Figure 4a. Figure 4b is deleted after our study.

Comments 18:  L276-285. Current form is good, but can you add IndVal for four different vegetations? It has the other potential to understand carabid communities.

Response 18: According to the expert's opinion, we add IndVal for four different vegetations in the table.

Comments 19:  L401-413. I agree that tundra is harsh environment, but it means that there are no trees providing stable habitats for large-bodied carabids. Wing loads, thin air, reduced oxygen and so on could be affected carabid distribution. However, sunlight without shading area is more important factor in my opinion, so tundra is like open-habitat in lower elevation. Open-habitats are generally frequently disturbed in terms of temperature and soil moisture that could be influenced by sunlight. Please carefully revise this paragraph.

Response 19: We fully agree with the experts. Wing loads, thin air, reduced oxygen could have an effect on insects, but the effect is very small. The main factors are probably temperature, light, length of growing season and food resources. We made major revision to this paragraph.

Comments 20:  L415. elevation gradien -> elevational gradient.

Response 20: According to the expert's opinion, we replaced “elevation gradien” with “elevational gradient”.

Comments 21:  L452. Insect flora -> Insect fauna

Response21: According to the expert's opinion, we replaced “Insect flora” with “Insect fauna”.

Reviewer 2 Report

Comments and Suggestions for Authors

The paper describes the carabid communities along an elevation gradient of about 2000 m in the Changbaishan Nature Reserve of NE China, the gradient covers three forest types of the temperate zone and, above 2000m, a tree-less zone that the authors call alpine tundra. The collecting method is undoubtely careful, and covers 16 elevation points, seven of which in the alpine belt, with 18,000 individuals collected, belonging to 62 species in two years sampling. Some parameters were examined in more detail, first of all the abundance and species diversity along the gradient, second the community characterization and, third, the body size and its variation along the coenocline. On the whole, the communities reflect an evident variation in relation to the altitude and the vegetation type, that remembers similar sequences observed in the European area, with a clear differentiation between the broadleaved mixed Korean pine forest and the conifer belt with spruce and fir, and a perhaps less defined birch belt that represents the boundary to the tree line in the sense of Koerner. The definition alpine tundra contains in fact two different vegetation types, a Vaccinium-Rhododendron belt and an more herbaceous culminal belt with Dryas and other alpine grasses and herbs which seems well identifiable in its indicator beetles, eg Carabus macleayi or Nebria pektusanica which represent the true specialistic inhabitants of the alpine prairie. Also in the sequence of the alpine altitudinal belts of Europe we observe, in fact, that the dwarf shrub belt with Rhododendron shares more affinity with the conifer belt and guests substantially a "forest community without forest", but the soils properties and the light conditions are more or less similar, but obviously the carabid species in Europe are different, eg Leistus nitidus and Calathus micropterus, often accompanied by some Pterostichus species. The interpretation of the communities is here made more complex by the effects of grazing, which destroys at least part of the Rhododendron cover and makes it patchy. The authors should emphasize this complexity of the "alpine tundra" and try to separate in the discussion the two types of D habitats, one rich of forest elements, the other really "true alpine" with the four indicators, the best being perhaps Pt. jeachi. I cannot give more suggestions about this, because also in the supplementary material the data are not presented in relation to the habitat sampled but only cumulated. Often also rare species can be of great indicator value, and for this reason I always suggest to present the raw data in the form of a "Zoosociological table" which gives a complete picture of the sampled species assemblages. some Amara species for example could help in the definition of the alpine communities (?).

The paper on the whole is worthy of attention, what is lacking is the examination of some other species traits very important in mountains, the dispersal power and the endemism, a milestone in all mountains.

I enclosed minor suggestions in the manuscript file, together with some orthography errors, and try to send in Choose File an extensive paper on the carabid assemblages of the Dolomites and another with the effects of climate change in the same area.

Comments on the Quality of English Language

on the enclosed manuscript

Author Response

The paper describes the carabid communities along an elevation gradient of about 2000 m in the Changbaishan Nature Reserve of NE China, the gradient covers three forest types of the temperate zone and, above 2000m, a tree-less zone that the authors call alpine tundra. The collecting method is undoubtely careful, and covers 16 elevation points, seven of which in the alpine belt, with 18,000 individuals collected, belonging to 62 species in two years sampling. Some parameters were examined in more detail, first of all the abundance and species diversity along the gradient, second the community characterization and, third, the body size and its variation along the coenocline. On the whole, the communities reflect an evident variation in relation to the altitude and the vegetation type, that remembers similar sequences observed in the European area, with a clear differentiation between the broadleaved mixed Korean pine forest and the conifer belt with spruce and fir, and a perhaps less defined birch belt that represents the boundary to the tree line in the sense of Koerner. The definition alpine tundra contains in fact two different vegetation types, a Vaccinium-Rhododendron belt and an more herbaceous culminal belt with Dryas and other alpine grasses and herbs which seems well identifiable in its indicator beetles, eg Carabus macleayi or Nebria pektusanica which represent the true specialistic inhabitants of the alpine prairie. Also in the sequence of the alpine altitudinal belts of Europe we observe, in fact, that the dwarf shrub belt with Rhododendron shares more affinity with the conifer belt and guests substantially a "forest community without forest", but the soils properties and the light conditions are more or less similar, but obviously the carabid species in Europe are different, eg Leistus nitidus and Calathus micropterus, often accompanied by some Pterostichus species. The interpretation of the communities is here made more complex by the effects of grazing, which destroys at least part of the Rhododendron cover and makes it patchy. The authors should emphasize this complexity of the "alpine tundra" and try to separate in the discussion the two types of D habitats, one rich of forest elements, the other really "true alpine" with the four indicators, the best being perhaps Pt. jeachi. I cannot give more suggestions about this, because also in the supplementary material the data are not presented in relation to the habitat sampled but only cumulated. Often also rare species can be of great indicator value, and for this reason I always suggest to present the raw data in the form of a "Zoosociological table" which gives a complete picture of the sampled species assemblages. some Amara species for example could help in the definition of the alpine communities (?).

The paper on the whole is worthy of attention, what is lacking is the examination of some other species traits very important in mountains, the dispersal power and the endemism, a milestone in all mountains.

I enclosed minor suggestions in the manuscript file, together with some orthography errors, and try to send in Choose File an extensive paper on the carabid assemblages of the Dolomites and another with the effects of climate change in the same area.

Response: We are very grateful to the expert for numerous meticulous and professional revision of the manuscript. It took him a lot of time and energy. We have carefully revised the paper according to the expert's opinion. Make the expression of the paper more clear. Changes made in the manuscript are highlighted in red. The author has answered the reviewer' comments one by one. The point-by-point response to the comments are highlighted in blue italics.

Comments 1: L51 Revise “insect physiology” to “physiology responses”.

Response 1: Response: We are very grateful to the expert for numerous meticulous and professional revision of the manuscript. It took him a lot of time and energy. According to the expert's opinion, we replaced “insect physiology” with “physiology responses”.

Comments 2: L58 Revise “becoming” to “may become”.

Response 2: According to the expert's opinion, we replaced “becoming” with “may become”.

Comments 3:  L 71 Revise “carabied” to “carabid”.

Response 3: According to the expert's opinion, we replaced “carabied” with “carabid”.

Comments 4:  L77 Revise “circle” to “cycle”.

Response 4: According to the expert's opinion, we replaced “circle” with “cycle”.

Comments 5:  L131-132 add “plant species”.

Response 5: According to the expert's opinion, we added “plant species” .

Comments 6:  L139 The tundra zone seems a less clear definition, the authors should give more detailson this landscape, e.g. if there are typical alpine grass mats or also dwarf shrubs.

Response 6: The tundra zone distributed above tree lines, with dwarf shrubs as the main vegetation community in the Changbaishan. We added these to the manuscript.

Comments 7:  L218-224 Revise “brich” to “birch”, place please (b) before "Richness and abundance..”.

Response 7: According to the expert's opinion, we replaced “brich” with “birch”. And placed (b) before "Richness and abundance”, and we also refer to the annotated form of previous papers in this journal.

Comments 8:  L 260 Revise “brich” to “birch”.

Response 8: According to the expert's opinion, we replaced “brich” with “birch”.

Comments 9:  L 338 in the Brenta Dolomites (Italy). There is no Mt. Albers in this country.

Response 9: We checked this part again, and the original handwriting was incorrect. We revised “Mt. Albers” to “Western Dolomites, Italy”.

Comments 10:  L 422-425 There is no clear relationship between size of ground beetles and duration of preimaginal development, small species like Trechus or Cymindis share winter larvae, and large Carabus are often quick spring breeders, also in alpine environments, eg Carabus creutzeri in Eastern Alps.

Response 10: I checked Saska, P. Temperature and development of central European species of Amara (Coleoptera: Carabidae). Eur. J. Entomol. 2003.Our expression was not rigorous enough, so we deleted this speculative conclusion.

Comments 11:  L 439 Revise “is” to “are”.

Response 11: According to the expert's opinion, we replaced “is” with “are”.

Comments 12:  L454 Revise “be” to “receive more”.

Response 12: According to the expert's opinion, we replaced “be” with “receive more”.

Comments 13:  L585 Revise “cylce” to “cycle”.

Response 13: According to the expert's opinion, we replaced “cylce” with “cycle”.

Reviewer 3 Report

Comments and Suggestions for Authors

General comments

The manuscript presents a study on the effect of elevation gradients on distribution and body size of carabid beetles in the Changbaishan Nature Reserve. Carabid beetles are studied in four different vegetation types, which are coniferous and broad-leaved mixed forest, spruce fir forest, birch forest and alpine tundra. The authors study if richness and abundance decrease with elevation, if body size increase with elevation and if these effects are affected by vegetation type. This is an interesting topic, which falls within the general scope of the journal. The manuscript is written in good English. The title well reflects its contents. The simple summary is understandable for non-specialists. The abstract is informative.

The introduction is well written, but I have doubts with respect to the text part regarding the Bergmann’s rule (see specific comment 4). Moreover, some information about the effect of food resources for carabid beetle larvae on body size of imagines is missing (see detailed comment 5). The research goals, described by three research questions, are well formulated. Material and methods are generally well described. The results chapter, however, needs substantial improvements and I also propose improvements in the discussion (see detailed comments). The Conclusions are generally well formulated.

The list of references contains a large number of 97 positions. The paper is supplemented by two tables, six figures and a supplemental data file, all of them important for the manuscript.

Detailed comments

1) Simple Summary, line 9: Instead of ‘This study will be first attempt…’ better ’This study will be a first attempt…’.

2) Simple Summary, lines 11-13: The sentence ‘Our found that the richness, abundance and body size of the carabid beetles linearly decrease with the elevation increase.’ seems to be incorrect at the beginning. Please check.

3) Introduction, line 77: Instead of ‘…is an important trait of insect…’ better ‘…is an important trait of insects…’.

4) Introduction, lines 82-91: I think it is necessary to be careful with the Bergmann’s rule when it comes to insects. While the Bergmann’s rule is considered to be valid for mammals, it is rather inconsistent in insects (e.g. Shelomi, 2012, The American Naturalist, Vol. 180, 511-519). Therefore, this text part has to be somewhat reformulated.

5) Introduction, lines 91-93: It is well know from publications (Ernsting et al., 1992, Ecological Entomology, 17, 33-42; Van Dijk, 1994, Ecological Entomology, 19, 263-270 – just to mention two of them) that size of carabid beetle imagines depends also on the food supply for the larvae. This aspect has to mentioned here and should be also taken stronger into account when discussing the results.

6) Introduction, lines 112-117: This text part repeats to some degree information already provided in lines 101-106.

7) Introduction, lines 119-124: I would propose a slight change in the order of the research questions. Aspects regarding body size are addressed both in the results and in the discussion at the end. Therefore I would mention the second research question as the last.

8) Material and Methods, Study area, line 127: Instead of ‘This study area was…’ better ’The study area was…’.

9) Material and Methods, Statistical analyses, lines 198-199: Which correlation coefficient was calculated (Pearson? Spearman?)? Please specify.

10) Results, Richness and abundance at different elevations, lines 209-211: Here the results are already discussed.

11) Results, Community composition, lines 248-251: The information provided here is very short. In my opinion, particularly from figure 4.a more information can be gained (for example, for me it seems that vegetation is more important than elevation, because it shows a somewhat better distinction along the first axis).

12) Results, Body size of species, lines 302-308: The results described in these lines do not fit to the results shown in figure 6. In the text significant correlations are described only for C. canaliculatus and C. venustus, and it is written that ‘There was no significant negative correlation between the body size of the other 13 species and elevation increase (Figure 6)’. However, in figure 6 I see significant negative correlations (p<0,05) for six species regarding the entire elevation, ten species regarding the elevations in forest habitat and two species in tundra habitat. It is clear from the R-values that the two correlations mentioned in the text as ‘significant’ are by far the strongest, but according to conventions in statistics other correlations with p<0.05 have also to be considered as significant. Therefore, this text part needs to be improved or it has to be specified convincingly in the methods, what is meant with ‘significant correlation’ in the manuscript.

13) Discussion, Body size of species and elevation, line 409: Instead of ‘…result may due to…’ better ‘‘…result may be due to…’.

14) Discussion, Body size of species and elevation, lines 425-429: The importance of food resources is only discussed very casually. However, food supply for larvae is of importance for body size of imagines in carabids (cf. detailed comment 5). It can be assumed that the food resources for the larvae of individual species change with elevation and/or habitat type. Therefore, this aspect has to be discussed more in detail.

15) Conclusions, lines 450-451: In my opinion it is redundant to say the body size is one of the factors which affects body size reduction. Therefore, I propose to modify this sentence.

Conclusion

The authors present a paper, which is interesting and scientifically important. Generally, the paper is well written, traceable and understandable. However, there are several flaws (see detailed comments). To conclude, I propose major revision of the manuscript before publication.

Comments on the Quality of English Language

I am not a native speaker, but in my opinion the manuscript is written in good English and needs only minor editing (see comments in the review).

Author Response

General comments

 The manuscript presents a study on the effect of elevation gradients on distribution and body size of carabid beetles in the Changbaishan Nature Reserve. Carabid beetles are studied in four different vegetation types, which are coniferous and broad-leaved mixed forest, spruce fir forest, birch forest and alpine tundra. The authors study if richness and abundance decrease with elevation, if body size increase with elevation and if these effects are affected by vegetation type. This is an interesting topic, which falls within the general scope of the journal. The manuscript is written in good English. The title well reflects its contents. The simple summary is understandable for non-specialists. The abstract is informative.

The introduction is well written, but I have doubts with respect to the text part regarding the Bergmann’s rule (see specific comment 4). Moreover, some information about the effect of food resources for carabid beetle larvae on body size of imagines is missing (see detailed comment 5). The research goals, described by three research questions, are well formulated. Material and methods are generally well described. The results chapter, however, needs substantial improvements and I also propose improvements in the discussion (see detailed comments). The Conclusions are generally well formulated.

The list of references contains a large number of 97 positions. The paper is supplemented by two tables, six figures and a supplemental data file, all of them important for the manuscript.

Detailed comments

Response: We are very grateful to the expert for numerous meticulous and professional revision of the manuscript. It took him a lot of time and energy. We are also very grateful for references provided by the expert. We have carefully revised the paper according to the expert's opinion. Make the expression of the paper more clear. Changes made in the manuscript are highlighted in red. The author has answered the reviewer' comments one by one. The point-by-point response to the comments are highlighted in blue italics.

Comments 1:Simple Summary, line 9: Instead of ‘This study will be first attempt…’ better ’This study will be a first attempt…’.

Response 1: According to the expert's opinion, we replaced “This study will be first attempt”with is “his study will be a first attempt”.

Comments 2: Simple Summary, lines 11-13: The sentence ‘Our found that the richness, abundance and body size of the carabid beetles linearly decrease with the elevation increase.’ seems to be incorrect at the beginning. Please check.

Response 2: According to the expert's opinion, we revised is sentence.

Comments 3: Introduction, line 77: Instead of ‘…is an important trait of insect…’ better ‘…is an important trait of insects…’.

Response 3: According to the expert's opinion, we replaced “is an important trait of insect” with “an important trait of insects”.

Comments 4: Introduction, lines 82-91: I think it is necessary to be careful with the Bergmann’s rule when it comes to insects. While the Bergmann’s rule is considered to be valid for mammals, it is rather inconsistent in insects (e.g. Shelomi, 2012, The American Naturalist, Vol. 180, 511-519). Therefore, this text part has to be somewhat reformulated.

Response 4: Thank you very much for the opinions of the experts, we have read this paper. Indeed, most of studies have shown insect often is inverse Bergman's rule pattern. We revised this section to make the expression more objective.

Comments 5: Introduction, lines 91-93: It is well know from publications (Ernsting et al., 1992, Ecological Entomology, 17, 33-42; Van Dijk, 1994, Ecological Entomology, 19, 263-270 – just to mention two of them) that size of carabid beetle imagines depends also on the food supply for the larvae. This aspect has to mentioned here and should be also taken stronger into account when discussing the results.

Response 5: Thanks very much for the expert's opinion, we did not find these papers when we started. After reading these papers, we added the content that temperature and food have a great influence on the development of larvae, and later adult size. We also added relevant content to the discussion.

Comments 6: Introduction, lines 112-117: This text part repeats to some degree information already provided in lines 101-106.

Response 6: According to the expert's opinion, we have modified these sentences and deleted some repetitive content to make the meaning more clear.

Comments 7: Introduction, lines 119-124: I would propose a slight change in the order of the research questions. Aspects regarding body size are addressed both in the results and in the discussion at the end. Therefore I would mention the second research question as the last.

Response 7: We agree with the expert's opinion, and we leave the second research question as the last.

Comments 8: Material and Methods, Study area, line 127: Instead of ‘This study area was…’ better ’The study area was…’.

Response 8: According to the expert's opinion, we replaced “This study”with is “The study”.

Comments 9: Material and Methods, Statistical analyses, lines 198-199: Which correlation coefficient was calculated (Pearson? Spearman?)? Please specify.

Response 9: It is Pearson’s correlation, and I added these in Materials and Methods. We use “r” instead “R” in the manuscript, including in the figure.

Comments 10: Results, Richness and abundance at different elevations, lines 209-211: Here the results are already discussed.

Response 10: According to the expert's opinion, we deleted “These data suggest that the variation of the vegetation type influences the correlation between richness and abundance and elevation.” in Results.

Comments 11:Results, Community composition, lines 248-251: The information provided here is very short. In my opinion, particularly from figure 4.a more information can be gained (for example, for me it seems that vegetation is more important than elevation, because it shows a somewhat better distinction along the first axis).

Response 11: We fully agree with the expert's opinion. Since there is some duplication between Figure 4a and Figure 4b, and we deleted Figure 4b. We added the content that “There were larger differences among the four vegetation types at different elevations”.

Comments 12: Results, Body size of species, lines 302-308: The results described in these lines do not fit to the results shown in figure 6. In the text significant correlations are described only for C. canaliculatus and C. venustus, and it is written that ‘There was no significant negative correlation between the body size of the other 13 species and elevation increase (Figure 6)’. However, in figure 6 I see significant negative correlations (p<0,05) for six species regarding the entire elevation, ten species regarding the elevations in forest habitat and two species in tundra habitat. It is clear from the R-values that the two correlations mentioned in the text as ‘significant’ are by far the strongest, but according to conventions in statistics other correlations with p<0.05 have also to be considered as significant. Therefore, this text part needs to be improved or it has to be specified convincingly in the methods, what is meant with ‘significant correlation’ in the manuscript.

Response 12: We fully agree with the expert's opinion. These statements in the original text are indeed incorrect, and we rewrote the content of this section. We use the size of the correlation coefficient value to illustrate the relationship between body size and elevation.

Comments 13:Discussion, Body size of species and elevation, line 409: Instead of ‘…result may due to…’ better ‘‘…result may be due to…’.

Response 13: During the revision process, we have removed this sentence.

Comments 14: Discussion, Body size of species and elevation, lines 425-429: The importance of food resources is only discussed very casually. However, food supply for larvae is of importance for body size of imagines in carabids (cf. detailed comment 5). It can be assumed that the food resources for the larvae of individual species change with elevation and/or habitat type. Therefore, this aspect has to be discussed more in detail.

Response 14: We also looked up relevant papers, and some studies pointed out that the lack of food resources in the larval stage will lead to smaller adult individuals. In the Changbaishan, the decrease in body size obviously occurs mainly in the alpine tundra zone, where the food resources are scarce and the feeding time is short. This is also the reason why individuals become smaller. We added these in the paper.

Comments 15: Conclusions, lines 450-451: In my opinion it is redundant to say the body size is one of the factors which affects body size reduction. Therefore, I propose to modify this sentence.

Response 15: This sentence really is not suitable in here. According to the expert's opinion, we deleted “The body size and distribution range of the carabid may be the factors that affect body reduction at high elevation.” in Conclusions.

Round 2

Reviewer 1 Report

Comments and Suggestions for Authors

You did great works. Thanks to your revision carefully.

Author Response

Comments and Suggestions for Authors

Comments 1: You did great works. Thanks to your revision carefully.

Response1: We are very grateful to the expert for his recognition of our research, and we have checked the full text and revised some language problems.

Reviewer 3 Report

Comments and Suggestions for Authors

This is the review if of the revised version of the manuscript. I congratulate the authors on a very good revision of the text. All my comments on the first version of the manuscript have been taken into account and the text has been improved accordingly.

However, I discovered only a few language errors, particularly in the new text parts (just two examples: in line 79 it should be ‘sometimes’ instead of ‘sometime’, in line 312 it should be ‘low’ instead of ‘lowe’). Therefore, I propose to publish manuscript after minor revisions (text editing).

Comments on the Quality of English Language

I discovered only a few language errors, particularly in the new text parts (just two examples: in line 79 it should be ‘sometimes’ instead of ‘sometime’, in line 312 it should be ‘low’ instead of ‘lowe’). Therefore, I recommend a final language check.

Author Response

Comments and Suggestions for Authors

Comments 1: This is the review if of the revised version of the manuscript. I congratulate the authors on a very good revision of the text. All my comments on the first version of the manuscript have been taken into account and the text has been improved accordingly.

However, I discovered only a few language errors, particularly in the new text parts (just two examples: in line 79 it should be ‘sometimes’ instead of ‘sometime’, in line 312 it should be ‘low’ instead of ‘lowe’). Therefore, I propose to publish manuscript after minor revisions (text editing).

Response1: We are very grateful to the expert for his recognition of our research. We fully accept expert advice, and we have checked the full text and revised some language problems. We replaced “sometime” with “sometimes”, and replaced “lowe” with “low”.

Comments on the Quality of English Language

Comments 2: I discovered only a few language errors, particularly in the new text parts (just two examples: in line 79 it should be ‘sometimes’ instead of ‘sometime’, in line 312 it should be ‘low’ instead of ‘lowe’). Therefore, I recommend a final language check.

Response2: We are very grateful to the expert for his recognition of our research. We fully accept expert advice, and we have checked the full text and revised some language problems. We replaced “sometime” with “sometimes”, and replaced “lowe” with “low”.